# Effect of Wheat Monoculture on Durum Wheat Yield under Rainfed Sub-Humid Mediterranean Climate of Tunisia

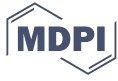

Asma Bouatrous [1,2,†], Kalthoum Harbaoui [1,3,*,†], Chahine Karmous [4], Samia Gargouri [5], Amir Souissi [1], Karima Belguesmi [6], Hatem Cheikh Mhamed [1], Mohamed Salah Gharbi [7] and Mohamed Annabi [1]

[1] Agronomic Sciences and Techniques Laboratory (LR16INRAT05), National Institute of Agronomic Research of Tunisia (INRAT), Carthage University, Hedi Karray Street, Ariana 2049, Tunisia; bouatrousasma@yahoo.fr (A.B.); souissiamir89@gmail.com (A.S.); hatemcheikh@yahoo.fr (H.C.M.); Mohamed.annebi@inrat.ucar.tn (M.A.)

[2] National Institute of Agronomy of Tunisia, Carthage University, 43 Avenue Charles Nicolle, Tunis 1082, Tunisia

[3] High School of Agriculture Mateur, Carthage University, Tabarka Road, Mateur 7030, Tunisia

[4] Laboratory of Genetics and Plant Breeding, National Institute of Agronomy of Tunis, 43 Avenue Charles Nicolle, Tunis 1082, Tunisia; karmouschahine@yahoo.fr

[5] Laboratory of Plant Protection (LR16INRAT04), National Institute of Agronomic Research of Tunisia (INRAT), University of Carthage, Hedi Karray Street, Ariana 2049, Tunisia; sgargouri90@gmail.com

[6] Regional Field Crops Research Center of Beja (CRRGC) BP 350, Béja 9000, Tunisia; k.belguesmi@yahoo.com

[7] Field Crops Laboratory (LR16INRAT02), National Institute of Agricultural Research of Tunisia (INRAT), Carthage University, Hedi Karray Street, Ariana 2049, Tunisia; gharbi.wheatpro@gmail.com

* Correspondence: harbaoui_kalthoum@yahoo.fr
† These authors contributed equally to this work.

**Abstract:** Cultivating cereals in monoculture systems contributes to the decrease in grain yield and quality. Currently, under Mediterranean climate conditions of Tunisia, wheat mono-cropping covers more than 70% of cereal areas. In order to reveal the impact of this practice on cereal productivity, five improved durum wheat cultivars (Karim, Khiar, Om Rabiaa, Razzek, and Maali) were conducted under two conditions of previous wheat crop: one-year wheat previous crop (W) and two successive years (W-W). Then, they were assessed for grain yield (GY), yield components (NKS, TKW, NS), straw yield, harvest index (SY, HI), and grain quality parameters during three consecutive cropping seasons (2017, 2018, and 2019). The results showed significant effects of cropping season for all measured parameters, except thousand kernel weight (TKW). A significant effect ($p < 0.05$) of Pre-Crop was observed on yield components. However, grain yield (GY) was improved after one-year wheat Pre-Crop (W) (4082.3 kg ha$^{-1}$) more than after two years (W-W) (3277.3 kg ha$^{-1}$). Our results show that, based on the three-year experiment, almost all yield related traits were significantly affected by the genotype except HI and NS. The highest GYs were recorded for Om Rabiaa (4010.4 kg ha$^{-1}$) and Nasr (3765.76 kg ha$^{-1}$). All grain quality was significantly ($p < 0.05$) affected by cropping season, but only gluten content (GC) and vitreousness aspect (Vit A) were affected by genotype. On the other hand, the Pre-Crop W-W decreased grain protein concentration (GPC) (12.13%) and GC (22.14%) but no significant effect was observed on the Vit A of grain in our study. Furthermore, GY was positively correlated with HI (r = 0.64), NKS (r = 0.59), SN (r = 0.49), GPC (r = 0.23), and GC (r = 0.23). According to stability analysis, the Karim cultivar is the most stable genotype in wheat mono-cropping for GY and straw yield (SY). Altogether, this study provides useful information for farmers on how to produce a satisfactory yield for durum wheat cultivation under mono-cropping wheat conditions in the sub-humid environment of the Mediterranean climate of Tunisia.

**Keywords:** durum wheat; grain yield; previous-crop; quality; stability

## 1. Introduction

Agriculture exists worldwide and allows farmers to grow and improve their crops with available inputs [1]. The agricultural sector plays a significant role for the path of economic development. It also contributes to the economic prosperity of many countries [2]. As a Mediterranean country, cereals in Tunisia are sown mainly under rain-fed conditions on about 1.5 million hectares, predominantly (about 60%) in the northern areas. Durum wheat (*Triticum turgidum* L. *var. durum*) is the major cereal species grown [3], due in part to its high selling price compared to bread wheat and other cereals [4]. Meanwhile, durum wheat productivity is highly variable from year to year (1 t ha$^{-1}$ to 6 t ha$^{-1}$) [5,6], closely linked to the variability and distribution of annual precipitation during the growing season as well as high temperatures during the grain filling stage [7]. The yield fluctuation of durum wheat is thought to continue and to worsen in the following year due to the climate change impacts in the Mediterranean area, especially Tunisia [8].

The use of improved cultivars and the adoption of appropriate crop management practices have significantly increased yields. Nevertheless, the average yield of 1.3 t ha$^{-1}$ is still not sufficient to meet increased consumer demand. Thus, adapting the widest agricultural practices has become urgent through the use of short-term rotations and monoculture.

In recent decades, continuous cereal mono−cropping replaced fallow [9]. In fact, wheat monoculture is common in several parts of Mediterranean countries such as Morocco, Syria, and Turkey [10]. Hence, cereal rotations with a large proportion of winter wheat are typical of large areas of Northern Europe and other humid climates [11], considering that winter wheat monoculture is recommended due to its economic impacts. However, mono−cropping contributes to losses in term of yield [11] and soil fertility [12]. On the other hand, cereal responses to cultivation under monoculture varies and so are limited by habitat conditions, agrotechnical measures used, and many other factors [13–15]. Furthermore, changes in the quality parameters of wheat grain are affected, most of all, by varietal traits, habitat conditions, cultivation system, and agrotechnical measures, including nitrogen fertilization [16–18]. Nitrogen is the major component of fertilizers which significantly influences crop yield and grain protein concentration [17].

Due to the above information, the basic task of the modern plant production is to strive for high, stable, and good-quality crops, with the lowest possible inputs and respect for the natural environment [19]. Durum wheat is an agronomically competitive crop to common wheat, which exhibits tolerance to biotic and abiotic stresses and is widely cultivated in low rainfall regions [20]. This crop is mainly cultivated in monoculture systems, especially in the north and northwest of Tunisia, and considered as a highly exigent agrosystem on nitrogen fertilizers. In fact, Tunisian farmers apply an average 150 kg ha$^{-1}$ of nitrogen annually. Nevertheless, the nitrogen use efficiency never overcomes 30% [21–23], which leads to huge problems either in soil quality or in environmental pollution.

Likewise, according to Rühlemann and Schmidtke [24], agriculture should not focus only on high crop yields, but must also consider the stable genotype. genotype stability has a pivotal role and simply means how consistent the yield of a genotype is compared with other ones [25]. However, Eberhart and Russell [26] proposed that genotypes with minimal interaction with the environment could be regarded as stable genotypes. Yield stability analysis relies on the assumption that linear correlation exists between growing conditions and genotypic performance [26,27]. The stability of cultivars was defined by high mean yield, regression coefficient (bi > 1), and a low deviation from the regression line (S2di) [28,29]. In this context, based on a three-year experiment, this research study aims to investigate the effect of continuous wheat monocropping on yield and yield components and grain quality of durum wheat as well as to evaluate different cultivars studied via stability analysis.

## 2. Material and Methods

### 2.1. Plant Material

Five durum wheat (*Triticum turgidum* L. *var. durum*) genotypes: Khiar, Karim, Maali, Nasr, and Om Rabiaa commonly sown in Tunisia were tested in this study. The characteristics of different genotypes are illustrated in Table 1 below.

**Table 1.** Origin, released date, and the main characteristics of the five genotypes of durum wheat included in the study.

| Genotypes | Breeder | Registration Date | Main Characteristics |
|---|---|---|---|
| Maali | INRAT | 2007 | Very productive cultivar (25% more than Karim), resistant to powdery mildew and fairly resistant to septoria and brown rust. More tolerant cutivar to drought than other cultivars of durum wheat. |
| Nasr | INRAT/ICARDA | 2004 | Productive, resistant cultivar to powdery mildew and yellow rust. |
| Karim | INRAT/CIMMYT | 1980 | Fairly resistant to septoria and brown rust. |
| Khiar | INRAT/CIMMYT | 1992 | Cultivars that are productive and relatively susceptible to brown rust and septoria diseases. |
| Om Rabiaa | INRAT/ICARDA | 1996 | |

### 2.2. Experimental Site

The experimental trials were conducted in field conditions over three cropping seasons, 2016–2017, 2017–2018, and 2018–2019, in the Field Crop Research Center at Beja, Tunisia (CRRGC) (36°43′32″ N; 9°10′54″ E; 248 m). The site is characterized by Mediterranean sub-humid climate with cold humid winters and very hot dry summers (Figure 1).

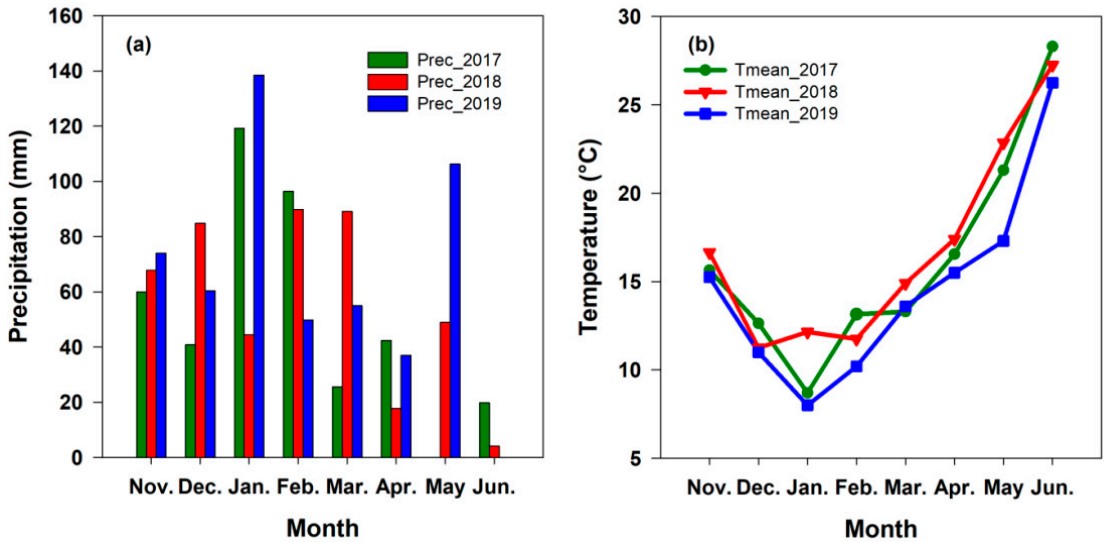

**Figure 1.** Climatic parameters of the experimental site measured during three cropping seasons: 2017, 2018, and 2019. (**a**) monthly precipitation pattern and (**b**) mean air temperature profile.

All assays were conducted on a silt clay loam soil texture (vertisol). Soil characteristics (organic matter, mineral nitrogen, and pH) during the three cropping seasons of study are illustrated in the following Table 2.

**Table 2.** Physicochemical properties of topsoil (0–40 cm) in Oued Beja station.

| Soil Properties | | Unit | * Mean Value |
|---|---|---|---|
| Clay (0.02–0.002 mm) | | % | 66 |
| Silt (0.2–0.02 mm) | | % | 23 |
| Sand (2.0–0.2 mm) | | % | 11 |
| pH | | | 7.2 |
| Organic matter | | % | 1.07 |
| | 2016–2017 | | 1183.24 |
| Mineral N | 2017–2018 | ppm | 1137.34 |
| | 2018–2019 | | 1055.40 |

*: mean values of three cropping seasons (2017, 2018, 2019).

### 2.3. Experimental Design and Management

The experiments were set out in a split plot design with four replicates. Five mainly sown cultivars, as described above, were sown under two previous crops: one-year wheat previous crop (W) and two-year wheat previous crop (W-W). For each cropping season, two blocks were divided into five subplots of 15 m$^2$ (5 m × 3 m), spaced by 1 m. During three cropping seasons (2017, 2018, and 2019), sowing was released at 1 December 2016, 25 November 2017, and 5 December 2018 using a conventional seeder at a seeding rate of 350 viable seeds/m$^2$. Three weeks before sowing, soil was ploughed at 20 cm depth and harrowed twice (10–15 cm depth).

The ammonium nitrate (NH$_4$NO$_3$; 33.5% N) was broadcast with 150 kg N ha$^{-1}$ at three phenological stages of durum wheat: 30% at beginning of tillering (GS24 of Zadoks scale, [30]), 40% at ear 1 cm (GS30 of Zadoks scale), and 30% at second node (GS32 of Zadoks scale).

Weeds were controlled by spraying mesosulfuron (30 g kg$^{-1}$), iodosulfuron (30 g kg$^{-1}$), and mefenpyrdiethyl (90 g kg$^{-1}$). The harvest was achieved when grain humidity was at 15%.

### 2.4. Yield and Yield Components

One-meter square in the middle of each experimental unit was harvested manually. Number of spike per meter square (SN), number of kernels per spike (NKS), thousand kernels weight (TKW), and grain yield (GY) were measured.

### 2.5. Straw Yield and Harvest Index

The straw yield (SY) was measured and the harvest index (HI) was calculated as the ratio of grain yield to biological yield.

### 2.6. Grain Quality Analysis

In order to measure humidity (Hr), grain protein concentrations (GPC), gluten content (GC), and vitreousness aspect of grain (VitA), a NIR grain analyzer (PERTEN Inframatic 9500, PerkinElmer, Waltham, MA, USA) was used as described in the ISO 16634-2:2009b Dumas method [31].

### 2.7. Statistical Analysis

The studied variables were analyzed using the MIXED procedure of SAS 9.0 [32]. Least significant differences (LSDs) for letter mean separation were assigned using the pdmix800 macro [33] with a significance level of 0.05. The mathematical model of the split-plot experiment is given by:

$$Y = \text{Cropping season} \mid \text{Wheat Pre-crop} \mid \text{genotype}$$

where Y = dependent variable (output variable).

A Pearson correlation test was performed between yield and quality traits. The JavaScript tool genotype Environment Analysis with R v 4.0.0 (GEA-R) [34] was used to

perform stability analysis in order to compare the different durum wheat genotypes based on regression coefficient (bi) and deviation from regression coefficient (S2di) [26].

## 3. Results

### 3.1. Yield Components as Affected by Wheat Pre-Crop

The statistical analysis of all yield parameters in the three years of experiment are shown in Table 3.

**Table 3.** Significance from ANOVA testing effect of cropping season (CS), wheat pre-crop (Pre-Crop) and genotype (G) and their interactions on spike number (SN), number of kernels per spike (NKS), thousand kernels weight (TKW), harvest index (HI), straw yield (SY), and grain yield (GY) of five durum wheat in the three cropping seasons. Data are averages ± standard errors.

| SV | SN | NKS | TKW (g) | HI | SY (kg ha$^{-1}$) | GY (kg ha$^{-1}$) |
|---|---|---|---|---|---|---|
| Cropping season (CS) | | | | | | |
| 2016–2017 | 278.95 ± 7.3 [a] | 32.044 ± 0.9 [a] | 47.47 ± 0.75 [a] | 0.49 ± 0.01 [a] | 4467.7 ± 222.07 [c] | 4278.5 ± 104.63 [a] |
| 2017–2018 | 303.25 ± 7.3 [a] | 28.54 ± 0.9 [b] | 47.40 ± 0.75 [a] | 0.38 ± 0.01 [b] | 6346.1 ± 222.07 [b] | 4012.4 ± 104.63 [a] |
| 2018–2019 | 250.9 ± 7.3 [b] | 24.86 ± 0.9 [c] | 45.76 ± 0.75 [a] | 0.26 ± 0.01 [c] | 7635.3 ± 222.07 [a] | 2748.5 ± 104.63 [b] |
| Wheat Pre-crop (Pre-Crop) | | | | | | |
| W-W | 247.18 ± 5.9 [b] | 25.40 ± 0.7 [b] | 48.11 ± 0.61 [a] | 0.39 ± 0.013 [a] | 5259.6 ± 81.32 [b] | 3277.3 ± 85.42 [b] |
| W | 308.22 ± 5.9 [a] | 31.57 ± 0.7 [a] | 45.63 ± 0.61 [b] | 0.37 ± 0.013 [a] | 7039.9 ± 81.32 [a] | 4082.3 ± 85.42 [a] |
| Genotype (G) | | | | | | |
| Om Rabiaa | 295.08 ± 9.42 [a] | 27.60 ± 1.16 [b] | 47.94 ± 0.97 [b] | 0.38 ± 0.016 [a] | 6679.6 ± 286.69 [a] | 4010.4a ± 135.07 [a] |
| Nasr | 285.76 ± 9.42 [a] | 29.89 ± 1.16 [a b] | 43.52 ± 0.97 [c] | 0.36 ± 0.016 [a] | 6744.7 ± 286.69 [a] | 3765.8a ± 135.07 [a b] |
| Maali | 275.67 ± 9.42 [a] | 27.01 ± 1.16 [b] | 50.93 ± 0.97 [a] | 0.35 ± 0.016 [a] | 6705.5 ± 286.69 [a] | 3608.7a ± 135.07 [b] |
| Khiar | 262.03 ± 9.42 [a] | 31.76 ± 1.16 [a] | 42.29 ± 0.97 [c] | 0.40 ± 0.016 [a] | 5330.0 ± 286.69 [b] | 3549.8a ± 135.07 [b] |
| Karim | 269.96 ± 9.42 [a] | 26.16 ± 1.16 [b] | 49.68 ± 0.97 [a b] | 0.39 ± 0.016 [a] | 5389.63 ± 286.69 [b] | 3464.4a ± 135.07 [b] |
| ANOVA | | | | | | |
| CS | *** | *** | NS | *** | *** | *** |
| Pre-Crop | *** | *** | ** | NS | *** | *** |
| CS × Pre-Crop | NS | NS | NS | NS | NS | *** |
| Genotype (G) | NS | ** | *** | NS | *** | * |
| CS × G | * | NS | NS | NS | NS | ** |
| Pre-Crop × G | NS | NS | NS | NS | ** | NS |
| CS × Pre-Crop × G | NS | NS | NS | NS | NS | ** |

Different lowercase letters indicate significant differences between all traits in each item ($p < 0.05$). ***: $p < 0.001$; **: $p < 0.01$; *: $p < 0.05$; NS: Not Significant.

The results showed a significant effect ($p < 0.001$) of cropping season (CS) on almost all yield traits except TKW. The lowest yield components' values were recorded during the third cropping season (2018–2019) for all traits except SY (Table 3). The highest values of GY, HI, SN, and NKS were obtained in the first and the second cropping seasons (2016–2017 and 2017–2018), while no significant differences were found for TKW between the first, second, and the third cropping season. Furthermore, the results showed that almost all yield-related traits were significantly affected by the genotype (G) except HI and NS. The variation in GY ranged from 3464.4 kg ha$^{-1}$ to 4010.4 kg ha$^{-1}$. The highest GYs were recorded for Om Rabiaa (4010.4 kg ha$^{-1}$) and Nasr (3765.76 kg ha$^{-1}$), but the lowest was obtained for Maali, Khiar, and Karim with an average of 3540.96 kg ha$^{-1}$. The highest NKS was recorded for Khiar cultivar, while the lowest value was observed for Karim. Besides, the genotypes Maali, Om Rabiaa, and Nasr showed the highest SY, with an average of 6709.93 kg ha$^{-1}$. Interestingly, the highest TKW (50.93 g) was recorded for Maali.

Previous crop significantly affects GY ($p < 0.001$), SY ($p < 0.001$), HI ($p < 0.001$), SN ($p < 0.001$), TKW ($p < 0.001$), and NKS ($p < 0.001$). However, it does not significantly affect HI and showed a significant 'CS × Pre-C' interaction for only GY. Indeed, GY (4082.3 kg ha$^{-1}$), SN (308.22), and NKS (31.57) were higher after one year of wheat Pre-Crop (W) than after two years of wheat Pre-Crop (W-W) (Table 3).

In the present study, despite the simple effects of Pre-Crop, G, CS, and GY were under the effects of the interaction between genotype and Pre-Crop. The highest values of GY were obtained for Om Rabiaa either after one year of wheat Pre-Crop (W) or after two years

of wheat Pre-Crop (W-W) (Figure 2), while a significant decrease in GY by 23.06% was observed between one year (W) and two years (W-W). Our investigation indicated that all genotypes in this study showed a decrease in GY after two years of wheat Pre-Crop (W-W).

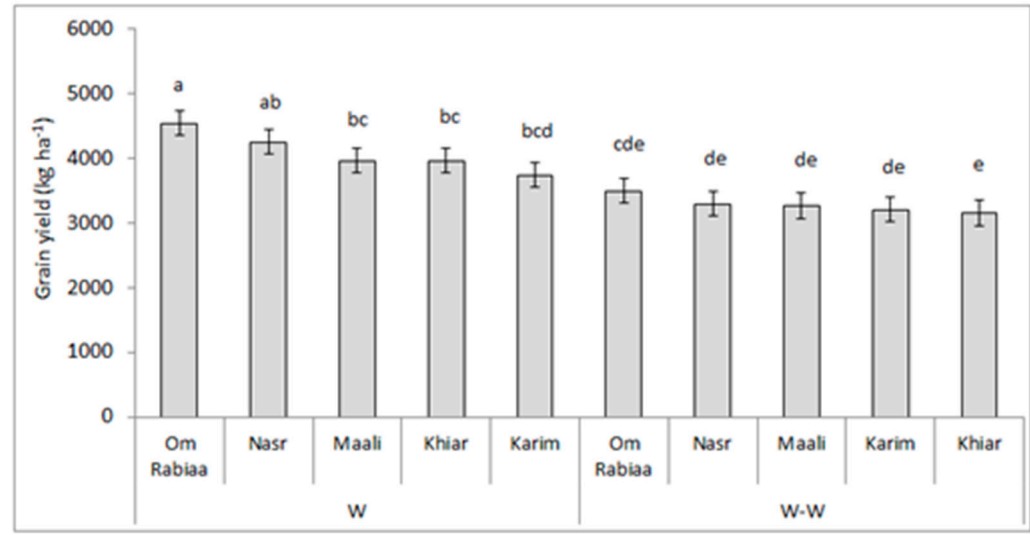

**Figure 2.** Effect of the wheat Pre-Crop × genotype on grain yield. Data are averages ± standard errors. Different lowercase letters indicate significant differences between all treatments in each item ($p < 0.05$).

### 3.2. Grain Quality Evaluation

The ANOVA showed that quality traits (humidity: Hr; grain protein concentration: GPC; gluten content: GC; vitreousness aspect: Vit A) were significantly affected by CS and G (Table 4). Interestingly, wheat Pre-Crop had a significant effect ($p < 0.05$) on gluten content (GC) and grain protein concentration (GPC) (Table 4).

**Table 4.** Significance from ANOVA testing effect of cropping season (CS), wheat pre-crop (Pre-Crop), and genotype (G), and their interactions on grain protein concentration (GPC), gluten content (GC), vitreousness aspect of grain (Vit A), and humidity (Hr) of five durum wheats during three cropping seasons.

| SV | GPC (%) | GC (%) | Vit A (%) | Hr (%) |
|---|---|---|---|---|
| Cropping season (CS) | | | | |
| 2016–2017 | 12.46 ± 0.14 [b] | 25.07 ± 0.28 [a] | 51.33 ± 1.19 [a] | 10.56 ± 0.04 [b] |
| 2017–2018 | 13.19 ± 0.14 [a] | 23.52 ± 0.28 [b] | 36.71 ± 1.19 [b] | 10.27 ± 0.04 [a] |
| 2018–2019 | 11.34 ± 0.14 [c] | 20.94 ± 0.28 [c] | 28.55 ± 1.19 [c] | 9.73 ± 0.04 [c] |
| Wheat Pre-crop (Pre-Crop) | | | | |
| W-W | 12.13 ± 0.11 [b] | 22.14 ± 0.23 [b] | 39.85 ± 0.97 [a] | 10.18 ± 0.03 [a] |
| W | 12.53 ± 0.11 [a] | 24.22 ± 0.23 [a] | 37.87 ± 0.97 [a] | 10.19 ± 0.03 [a] |
| Genotype (G) | | | | |
| Om Rabiaa | 12.70 ± 0.18 [a] | 24.00 ± 0.36 [a] | 35.41 ± 1.69 [c] | 10.25 ± 0.05 [a] |
| Nasr | 12.46 ± 0.18 [a] | 23.55 ± 0.36 [a b] | 42.47 ± 1.69 [b] | 10.12 ± 0.05 [a] |
| Karim | 12.26 ± 0.18 [a] | 23.05 ± 0.36 [a b] | 47.02 ± 1.69 [a] | 10.25 ± 0.05 [a] |
| Khiar | 12.11 ± 0.18 [a] | 22.65 ± 0.36 [b] | 33.00 ± 1.69 [c] | 10.18 ± 0.05 [a] |
| Maali | 12.10 ± 0.18 [a] | 22.63 ± 0.36 [b] | 36.40 ± 1.69 [c] | 10.14 ± 0.05 [a] |

**Table 4.** *Cont.*

| SV | GPC (%) | GC (%) | Vit A (%) | Hr (%) |
|---|---|---|---|---|
| ANOVA | | | | |
| CS | *** | *** | *** | *** |
| Pre-Crop | * | *** | NS | NS |
| CS × Pre-Crop | NS | NS | NS | NS |
| Genotype (G) | NS | * | *** | NS |
| CS × G | NS | NS | NS | NS |
| Pre-Crop × G | NS | * | ** | NS |
| CS × Pre-Crop × G | NS | NS | NS | NS |

Different lowercase letters indicate significant differences between all traits in each item ($p < 0.05$). ***: $p < 0.001$; **: $p < 0.01$; *: $p < 0.05$; NS: $p \geq 0.05$.

Moreover, 'Pre-Crop × G' interaction showed a significant effect on GC and Vit A of grain. The highest GPC (13.19%) and gluten (25.07%) values were obtained during the second cropping season, 2017–2018, whereas the lowest values of the same parameters were registered at the third cropping season (2018–2019). As for gluten content, means values ranged from 22.6% for Maali to 24% for Om Rabiaa (Table 4).

Furthermore, GPC parameter was significantly ($p < 0.01$) affected by Pre-Crop, which decreased from 12.53% under one-year wheat Pre-Crop (W) to 12.13% under two years wheat Pre-Crop (W-W). Meanwhile, a significant ($p = 0.001$) decrease in gluten content by 8.58% was observed between one year (W) and two years (W-W) wheat pre-cropping.

In addition, almost all grain quality parameters were slightly affected by genotype, except vitreousness aspect (Vit A), which was significantly related to genotypic variability. Indeed, the highest value of Vit A (47.02%) was observed for Karim, whereas the lowest value of Vit A (33.00%) was registered for Khiar (Table 4).

### 3.3. Relationship between Yield, Agronomic Parameters, and Grain Quality Traits

Correlations between GY and yield components were evaluated for the five durum wheat cultivars tested in this study. Interestingly, GY was positively correlated with the HI (r = 0.64), NKS (r = 0.59), SN (r = 0.49), GPC (r = 0.23), and gluten content (r = 0.23) (Table 5). In addition, significant correlations were also observed between SN and both GPC (r = 0.31) and gluten (r = 0.41). Moreover, TKW was negatively correlated with NKS (r = −0.36) and positively correlated with Vit A (r = 0.20). The results also showed that SY was significantly correlated with SN (r = 0.21) and negatively correlated with HI (r = −0.82) and Vit A (r = −0.46).

**Table 5.** Pearson correlation matrix between yield components: grain yield (GY, kg ha$^{-1}$), straw yield (SY, kg ha$^{-1}$), harvest index (HI), spike number (SN), thousand kernels weight (TKW, g), number of kernels per spike (NKS), and grain quality traits: humidity (Hr, %), grain protein concentration (GPC, %), gluten content (GC, %), vitreousness aspect of grain (Vit A, %).

| | GY | SY | HI | SN | TKW | NKS | Hr | GPC | GC | Vit A |
|---|---|---|---|---|---|---|---|---|---|---|
| GY | 1 | | | | | | | | | |
| SY | −0.24 ** | 1 | | | | | | | | |
| HI | 0.68 *** | −0.82 *** | 1 | | | | | | | |
| SN | 0.52 *** | 0.21 * | 0.08 | 1 | | | | | | |
| TKW | −0.06 | −0.11 | 0.04 | −0.05 | 1 | | | | | |
| NKS | 0.39 *** | −0.10 | 0.31 *** | 0.10 | −0.36 *** | 1 | | | | |
| Hr | 0.43 *** | −0.28 *** | 0.42 ** | 0.24 ** | 0.13 | 0.20 * | 1 | | | |
| GPC | 0.47 ** | −0.15 | 0.33 * | 0.31 *** | 0.04 | 0.17 | 0.49 *** | 1 | | |
| Gluten | 0.50 *** | 0.02 | 0.20 * | 0.41 *** | 0.02 | 0.26 ** | 0.49 *** | 0.88 *** | 1 | |
| Vit A | 0.35 *** | −0.46 *** | 0.53 *** | 0.08 | 0.20 * | 0.20 * | 0.31 *** | −0.02 | 0.02 | 1 |

***: $p < 0.001$; **: $p < 0.01$; *: $p < 0.05$; NS: $p \geq 0.05$.

### 3.4. Stability Analysis

The average of GY and SY were quite stable across cropping seasons and were highly affected by Pre-Crop and CS. GY showed large variability within the tested durum wheat genotypes. GY and SY stability are considered as important yield components mainly under mono-cropping. The stability parameters were determined according to Eberhart and Russell [27] (Table 6).

**Table 6.** Estimation of mean performance, regression coefficient (bi), and deviation from regression coefficient (S2di) for grain yield (GY) and straw yield (SY) parameters.

| Genotypes | GY | | | SY | | |
|---|---|---|---|---|---|---|
| | **Means** | **bi** | **S2di** | **Means** | **bi** | **S2di** |
| Karim | 3464.37 | 1.12 | −120,578.20 | 5388.95 | 0.86 | −581,796.45 |
| Khiar | 3549.75 | 0.62 | 271,800.82 | 5329.97 | 1.13 | −277,370.26 |
| Maali | 3608.70 | 1.07 | 32,331.11 | 6705.45 | 0.64 | −459,455.14 |
| Nasr | 3765.76 | 1.24 | 66,790.68 | 6644.65 | 1.28 | −444,358.00 |
| Om Rabiaa | 3010.37 | 0.92 | 209,702.11 | 6679.62 | 1.06 | −586,481.72 |

The variations in bi values indicated that the tested genotypes responded differently to Pre-Crop in different cropping seasons. The bi of SY ranged from 0.8 to 1.28. However, the bi of GY ranged from 0.6 to 1.24. Thus, high bi, mean of GY, SY, and slow deviation are required for stable genotypes. According to those considerations, Karim showed a comparatively stable response with GY mean (3464.37 kg ha$^{-1}$) and least mean square deviation (S2di = −120,578.20). Integrating SY as an important agronomical component with GY for stability analysis showed that the genotypes combining high mean, bi, and low S2di were Om Rabiaa and Karim. Therefore, Karim showed the best stability for GY and SY in the wheat mono-cropping conditions. This genotype had a high average of GY of 3464.37 kg ha$^{-1}$ (bi = 1.12; S2di = −120,578.20) and high SY of 5388.95 kg ha$^{-1}$ (bi = 0.86; S2di = −581,796.45) (Figure 3).

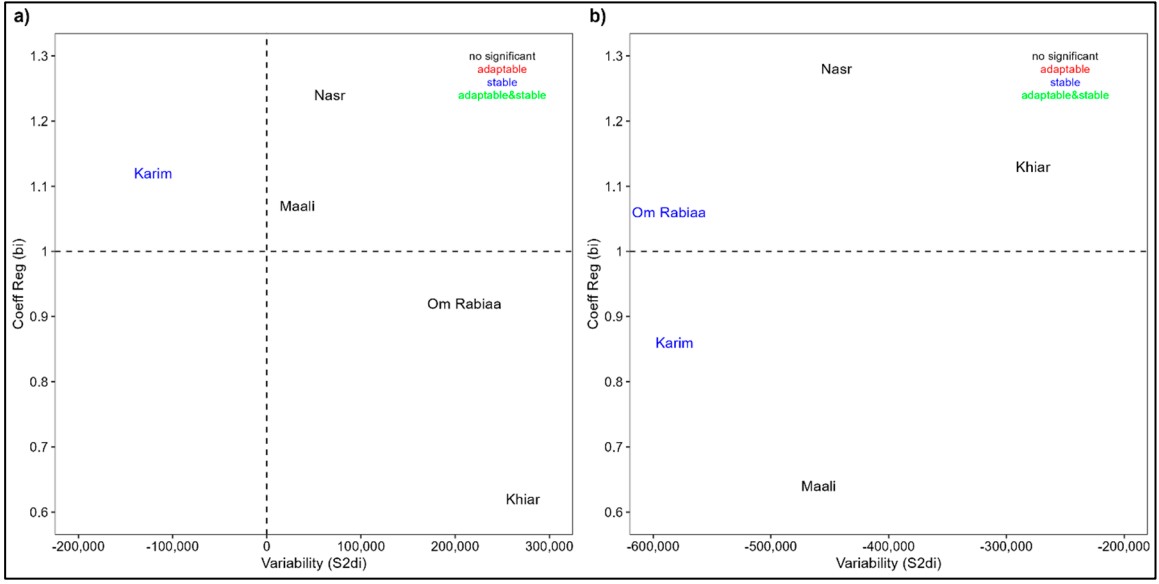

**Figure 3.** Stability analysis of GY (**a**), SY (**b**), coefficient of variability (X-axis), and coefficient of regression (Y-axis).

## 4. Discussion

### 4.1. Agronomic Performance

Given that in Tunisia, cereal farming, especially durum wheat, is mainly rainfed [35], precipitation and mean ambient temperature are the most important climatic factors that affect its productivity. In the present study, variation in grain yield between the three growing seasons was likely due to the rainfall distribution irregularity, especially during sensitive stages. Interestingly, at the two first years (2017 and 2018) of the assay, the rainfall pattern was more favorable to improve yield than the same pattern in the third year (2019), which was negatively affecting crop yield. In this last season crop, the lack of sufficient water during the growth stage from February to April along with relatively high temperatures explained the behavior of plant development and, therefore, productivity. In our study, the good rainfall distribution during the wheat development stage in 2017 and 2018 was probably leading to good water availability for the plant, especially during the elongation phase (from February to April). Many authors [36,37] reported that the rainfall distribution during the growing season could greatly affect durum wheat grain yield. Our results showed that water availability in the first two years of the study (2017 and 2018) was positively affecting grain yield compared to the last year (2019). Ercoli et al. [38] reported a direct relationship between grain yield and rainfall distribution during the reproductive phase. Variability of cereal yields in the Mediterranean basin is mainly attributed to inadequate and erratic seasonal rainfall [39]. In this context, under Mediterranean environments, the supplemental irrigation of cereals during the reproductive and grain filling growth stage can contribute to alleviating yield reduction caused by drought [37,40,41]. Indeed, it was usually confirmed that durum wheat grain yield fluctuation in Mediterranean regions was known as a frequent issue [24].

The previous crop is considered as important information to know before installing a cereal crop. Indeed, it plays a major role in increasing or decreasing cereal production because of this crop's high need for the nitrogen left in the soil by good previous crops. Elsewhere, our study demonstrated that GY was significantly affected by Pre-Crop, which was reduced to about 20% under successive cereal monoculture. Indeed, GY was varied from 4082.3 kg ha$^{-1}$ under one year Pre-Crop (W) to 3277.3 kg ha$^{-1}$ under two years Pre-crop (W-W). This variation was due to a lower SN and NKS after second condition (W-W) than after first condition (W). Our findings are in agreement with Woźniak [42], who reported that winter wheat sown in the 29-year crop monoculture produced 32% grain yield, lower than that grown in the crop rotation system, and this grain yield reduction was due to a low number of spikes per m$^2$, short spikes, low grain weight per spike, and low TKW. The same authors demonstrated that wheat growth is affected by cereal monoculture compared to a crop rotation system. Similar observations were reported by other authors [43,44].

Moreover, genotypic variability could be involved in the adaptation levels of genotypes to different previous crop conditions. The current assay is proving the impact of genotypic variability on durum wheat production. Interestingly, genotypic variability was observed for GY, TKW, NKS, and SY among tested durum wheat genotypes. GY genotypic variability has been largely reported in wheat and other cereals [45–48]. Furthermore, our results demonstrated that TKW is highly affected by genotype, which is in agreement with Arduini et al. [49]. The improvement in TKW might be attributed to a better nutritional durum wheat status and, thus, higher grain filling and development [50]. Moreover, our result showed a positive correlation between GY, SN, and NKS. In fact, GY improvement was mainly due to the increase in spikes and kernels per spike [51]. Positive correlation between GY and HI could be explained by the fact that genotypes with higher HI tended to improve their GY. These findings confirmed that Karim, known as a short cultivar, has a high GY potential and low SY, which is in agreement with other researchers [35]. Based on the stability analysis, the same cultivar was classified as the most stable genotype under successive monocropping conditions compared with other tested genotypes. This finding is in agreement with El Felah and collaborators [52].

*4.2. Grain Quality Evaluation*

As already noted for yield and agronomic traits, grain quality parameters were also greatly affected by climatic conditions. This result agrees with the findings of Mariani et al. [53] and Ames et al. [54]. Through this analysis, we demonstrated that grain protein concentration (GPC) of durum wheat grain is relatively dependent on the Pre-Crop and weather conditions. The highly significant effect of Pre-Crop could be explained by the fact that this quality trait is not only genetically inherited, but also significantly modified by agronomic practices and environmental factors, as reported by Campillo et al. [55]. However, the mean grain vitreousness aspect (Vit A) ranging from 33.00% for Khiar to 47.02% for Karim was highly affected either by genotype or season crop (CS) and then by the 'Pre-Crop × Genotype' interaction. In the same context, previous studies have also indicated high variability of the vitreousness aspect between Syrian durum wheat cultivars [56]. Some investigators explained the large variation in the Vit A parameter between genotypes and 'Genotype × Location' effects in the drylands [57]. Our results showed that tested cultivars responded differently during the three-cropping seasons. Beyond genotypes, the mean of GC ranged from 22.63% for Maali to 24.00% for Om Rabiaa cultivars. These results are in agreement with others' reports [58].

**5. Conclusions**

This study showed that all yield traits, except TKW, of durum wheat tested cultivars were affected by wheat Pre-Crop (W-W and W), while only some quality traits (GPC and GC) were affected by this variable factor (Pre-Crop). Therefore, wheat monocropping, largely practiced in the north and northwest of Tunisia, is considered as a real problem for durum wheat production. Our results could be useful to understand the behavior of the genotypes used under successive wheat monoculture. Then, our findings highlight stability among five durum wheat genotypes for GY and SY, and lead us to identify that the Karim genotype could be the most stable one in wheat mono-cropping conditions in a sub-humid environment. The agronomic interest of this research is to release which genotype could be the best in this specific monoculture condition of Tunisian cereal regions, and we proved that Karim could be this candidate. Finally, it is important that medium to long term studies on wheat monocropping cultivation are conducted to improve the references and better guide local farmers towards a better yield.

**Author Contributions:** Conceptualization, K.H. and M.S.G.; Data curation, A.B., K.H., S.G., and K.B.; Formal analysis, A.B. and H.C.M.; Funding acquisition, K.H. and M.A.; Investigation, A.B.; Methodology, K.H., K.B., and M.S.G.; Project administration, M.A.; Resources, M.A.; Software, A.B. and A.S.; Supervision, K.H. and M.A.; Validation, M.A.; Writing—original draft, A.B. and K.H.; Writing—review and editing, C.K. All authors have read and agreed to the published version of the manuscript.

**Funding:** This study was supported by the Agronomic Sciences and Techniques Laboratory (LR16INRAT05) of the National Institute of Agronomic Research of Tunisia (INRAT), ChangeUp: 'Innovative Agro-Ecological Approaches to Achieving Resilience to Climate Change in Mediterranean Countries', funded by Tunisian Higher Education and the Scientific Research Ministry (PRIMA-Med Call 2021 Section 2).

**Conflicts of Interest:** The authors declare no conflict of interest.

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
