# Peer review of "Effect of Wheat Monoculture on Durum Wheat Yield under Rainfed Sub-Humid Mediterranean Climate of Tunisia"

_agronomy, doi:10.3390/agronomy12061453_

Round 1
Reviewer 1 Report
The manuscript submitted by AsmaBouatrous et al. demonstrated the Effect of Successive Cereal Monoculture previous crop on durum wheat (Triticumturgidum L. var. Durum) Yield and Quality under rainfed sub humid Mediterranean Climate.The article is sufficiently clear. English is good and comprehensible, even though it is suggested to perform an accurate revision for the presence of a few grammar inaccuracies, and/or residues of copy/paste actions. The writing style should be a little improved for some aspects, and at this regard there are additional suggestions for the authors in the section of the general comments.
The manuscript submitted by Asma Bouatrous et al. demonstrated the Effect of Successive Cereal Monoculture previous crop on durum wheat (Triticum turgidum L. var. Durum) Yield and Quality under rainfed sub humid Mediterranean Climate. The article is sufficiently clear. English is good and comprehensible, even though it is suggested to perform an accurate revision for the presence of a few grammar inaccuracies, and/or residues of copy/paste actions. The writing style should be a little improved for some aspects, and at this regard there are additional suggestions for the authors in the section of the general comments.
Line 32- Replace “ha-1” by “ha-1” and check throughout the manuscript .
Line 32-33. This sentence have no sense “Elsewhere, grain quality parameters were evaluated including grain protein content (GPC), gluten content (GC) and vitreousness aspect (Vit A) of grain.” Write its increase or decrease due to monoculture practices.
Line 92-97. The objectives of the study is not clear.
Table 3. Add units in the tables
- The title of article should be improved and attractive.
- In the abstract the authors focused to write what they assessed in the experiment, but they did not clarify the results, likewise the improvement in yield components etc.
- Add a little introduction on durum wheat, fertilizer requirements, its production etc
- Separate the section of the methodology of growth yield and yield components.
- Table 2; and Table 3; add units to the each indicator. If the same unit is using for indicator then separated in column. Abbreviation term in the table caption is not appropriate, write Note; and then mention your captions with units etc
- Make the conclusion short and attractive.
- Check the references and revise it according to journal format.
Author Response
Please see the attachement

Reviewer 2 Report
Topical research issues with a practical aspect. Purpose and research methods correct. Relevant statistics. Selection of literature and correct citation. The article is suitable for publication
Author Response
There is no comments to answer
This manuscript is a resubmission of an earlier submission. The following is a list of the peer review reports and author responses from that submission.
Round 1
Reviewer 1 Report
Although a three-year field experiment, the manuscript was too simple to investigate why wheat mono-cropping systems affects grain yield and quality of durum wheat. Maybe it is more suitable for local Journals.
- Lines 48-51 “Meanwhile, durum wheat productivity is characterized by high inter-annual variability (1 t.ha−1to6 t.ha−1) attributed to poor crop management [6] and drought stress [7] linked to rainfall annual and seasonal fluctuations [4] as 50 well as high temperatures during grain filling stages [8].” should be rephrased.
- Lines 59-60, “Therefore, mono-cropping cereal systems could reduce productivity in dry areas of 59 the Mediterranean region.” What do you mean “therefore”, I could not understand the relationships between above-mentioned sentences.
- Authors should re-arrange paragraphs in Lines 69-90.
- Why data of SN across three cropping seasons was not equal to that across genotypes, as well as for SY?
Reviewer 2 Report
There is clarity in the approach to the problem, as well as in the results that are sought to examine the effects of wheat mono-cropping systems on yield and quality performance of five most cultivated durum wheat genotypes under sub humid Mediterranean environment.
The method and experimental design are clear and show innovation on the implementation and analysis of short-term wheat monoculture, to improve wheat productivity under various agricultural strategies.
Abstract
The abstract does not clearly describe the data analysis method used in relation to the results obtained.
Line 24
Either remove or relocate the citation to another part of the text. The abstract should not contain any citations.
Line 24
Check spelling. “In orders…”
Line 23-24
Clarify the idea in expression "cultivated species and even cultivar."
Line 27
Review the use of uppercase and lowercase letters. “Crop.”
Line 30
Check spelling. “Thousand kernels weight.”
Line 37
What is "SY"? There is no description.
Line 68
Check spelling. “decreaseofgrain.”
Line 88
Check space. “In particular.”
Line 22
Check space. “Straw yield (SY).”
Line 242-244
The objective of the study should also be specified in the section corresponding to the introduction.
Line 267
The hypothesis of the study should also be specified in the section corresponding to the introduction.
Reviewer 3 Report
The research article entitled “Effect of Successive Cereal Monoculture previous crop on durum wheat (Triticum turgidum L. var. Durum) Yield and Quality under a rainfed sub-humid environment in Tunisia” reports very interesting experimentation led aim of the three-consecutive cropping season. This study aimed to evaluate the yield and quality of durum wheat in wheat monoculture conditions. In particular, the field trial was conducted to assess the variability of durum wheat yield components and grain quality of commonly used genotypes under continuous cereal cropping rotation in sub-humid conditions.
The article is sufficiently clear. English is good and comprehensible. The subject of this work is interesting, the methods used in the study are acceptable and the analysis and discussions of the results are reasonable. But after review of this draft, I found that this manuscript has serious flaws, which need to be addressed before its acceptance for publication. I suggest that authors to must consider the following points to make it publishable in the journal of Agronomy.
Major comments:
1. I suggest adding soil physiochemical properties data of each year in a table to understand clearly the impact of these varieties on soil quality or to know the effect of soil properties on varieties in each year.
2. Table 3 and Table 4. The authors did a big mistake in analyzing the data in this table. All the SE values are the same in the table. The authors need to re-analyze the data and write the ±value (STDEV or Standard error) correctly. Furthermore, the table title should be on top and the caption of the table should be moved below. The authors also did not mention that what does ±value indicate?
3. Figure 2. What does the bar mean above the columns? Is it a standard error? If yes then why do all standard errors look the same? Is it possible to have the same SE value for all treatments?
(4) In table 5 separate the title and captions, move the captions below the table. Check throughout the tables.
Other comments
- The font style and size are not the same in the abstract.
- In abstract. The author mentioned that “Grain quality parameters were evaluated including grain protein content (GPC), gluten content (GC) and vitreousness aspect (Vit A) of grain’. But did not mention its responses to different seasons and verities.
- In line 29 the abbreviation CS is used for the 1st time in the abstract, it should be explained.
- In abstract, the authors mentioned that “The Pre-Crop W-W decreased GPC (12.13%) and GC (22.14%)”, but the authors did not mentioned that this decrease was compared to varieties or season, if its season then what about these values, are these values are the average of all verities. The overall abstract need to improve, such as providing a percent increase/decrease to each season/year or varieties and should be compared to the previous season and other varieties.
- line 51-52, this sentence looks incomplete, is there only climatic change reason to influence wheat fluctuation.
- Line 118. Remove ‘….’ From the sentence. Check throughout the manuscript.
- Section 2.5, add a reference.
- Section 2.5, add references or write the full procedure of this methodology.
- Line 256, and line 292 correct the reference style.
- Line 292-296, which results (improved or decreased their parameters) are in agreement? The authors just mentioned that the agronomic traits, grain quality parameters were also greatly affected by climatic conditions.
References:
- 1st of all, all the citations seemed to be a not recent citations, the autours should provide or change the updated citation in recent 5 to 10 years.
- Reference 10. There is only one citation of Jones in this article i.e. “Jones, M.J.; Singh, M. Time-trends of yield in long-term trials. Exp. Agric. 2000b, 36, 165–179” then why the authors used “b” letter with the year.
- Why reference number 53 is in red color?
Recheck all the references and correct it according to the journal format.
Round 2
Reviewer 1 Report
Although the authors revised following some suggestions of previous comments, I have to reject it again. As stated above, the paper only paid attention on grain yield and quality, but information on the agronomic and physiological traits is still lacking. I think this paper is suitable for local Journals.